# Dissipative optomechanics in high-frequency nanomechanical resonators

André G. Primo[1,4], Pedro V. Pinho [1,4], Rodrigo Benevides [2], Simon Gröblacher [3], Gustavo S. Wiederhecker[1] & Thiago P. Mayer Alegre [1]✉

The coherent transduction of information between microwave and optical domains is a fundamental building block for future quantum networks. A promising way to bridge these widely different frequencies is using high-frequency nanomechanical resonators interacting with low-loss optical modes. State-of-the-art optomechanical devices rely on purely dispersive interactions that are enhanced by a large photon population in the cavity. Additionally, one could use dissipative optomechanics, where photons can be scattered directly from a waveguide into a resonator hence increasing the degree of control of the acousto-optic interplay. Hitherto, such dissipative optomechanical interaction was only demonstrated at low mechanical frequencies, precluding prominent applications such as the quantum state transfer between photonic and phononic domains. Here, we show the first dissipative optomechanical system operating in the sideband-resolved regime, where the mechanical frequency is larger than the optical linewidth. Exploring this unprecedented regime, we demonstrate the impact of dissipative optomechanical coupling in reshaping both mechanical and optical spectra. Our figures represent a two-order-of-magnitude leap in the mechanical frequency and a tenfold increase in the dissipative optomechanical coupling rate compared to previous works. Further advances could enable the individual addressing of mechanical modes and help mitigate optical nonlinearities and absorption in optomechanical devices.

The burgeoning field of cavity optomechanics combines the reliability of long-range information transport using optical photons with the versatility of nanomechanical oscillators. This conjunction enabled a plethora of demonstrations, including high-precision force and displacement sensors[1-3], and the synchronization of mechanical oscillators[4,5] for signal processing. Furthermore, in high-frequency optomechanical systems, it is possible to actively control the strength of the creation or annihilation scattering process of long-lived phonons[6-9]. This property allows the quantum control of optomechanical systems with promising applications in coherent quantum microwave-to-optical conversion[10-19], and quantum memories[20,21].

In standard dispersive optomechanical devices, the acoustic modes are engineered to shift the optical cavity resonance frequency, $\omega_0$[22-24]. As a consequence, only photons confined to the resonator field are efficiently scattered. This mechanism is quantified by the dispersive optomechanical frequency pulling, $G_\omega = -\frac{d\omega_0}{dx}$, where $x$ is the mechanical displacement amplitude of a given acoustic mode. In dissipatively coupled optomechanical systems, however, the acousto-optic interaction may take place between the optical excitation channel and the cavity[25-28], e.g., through a mechanical modulation of the bus waveguide-cavity coupling rate $\kappa_e$. Its strength is quantified by the dissipative pulling $G_{\kappa_e} = \frac{d\kappa_e}{dx}$. In these systems, photons may be

[1]Gleb Wataghin Institute of Physics, University of Campinas, 13083-859 Campinas, SP, Brazil. [2]Department of Physics, ETH Zürich, 8093 Zürich, Switzerland. [3]Kavli Institute of Nanoscience, Department of Quantum Nanoscience, Delft University of Technology, Lorentzweg 1, 2628CJ Delft, The Netherlands. [4]These authors contributed equally: André G. Primo, Pedro V. Pinho. ✉e-mail: alegre@unicamp.br

scattered directly from the waveguide into the cavity mode, as illustrated in Fig. 1a.

In conjunction with its dispersive counterpart, the presence of the dissipative scattering mechanism leads to remarkable interference phenomena in both mechanical and optical spectra. For instance, the interplay between dispersive and dissipative couplings can vary if a given mechanically induced shift in the optical frequency is accompanied by either a (mechanically induced) increase or decrease in optical linewidth, allowing a yet-to-be-explored tool to control optomechanical interactions. Furthermore, using dissipative coupling one could achieve the optomechanical ground-state cooling even in the bad-cavity limit[25,29], where the mechanical frequency is smaller than the total optical linewidth, $\Omega \ll \kappa$, which is unfeasible using dispersive optomechanics alone. Similar results could be obtained in more complex systems, e.g. using Fano optical lineshapes in hybrid atom-optomechanical systems for sideband suppression/enhancement[30]. Although simpler, the dissipative optomechanics approach requires small intrinsic losses, $\Omega \gg \kappa_i$, a regime thus far elusive[1,31–33], hence inhibiting the application of dissipative optomechanics to coherent information swap protocols[6]. State-of-the-art devices still need an order-of-magnitude leap in mechanical frequencies to reach this regime, while preserving low optical losses.

Here, we demonstrate an optomechanical system in the sideband-resolved regime ($\Omega/\kappa \approx 10$) which displays both dissipative and dispersive optomechanical couplings. Our mechanical modes, operating at $\Omega/(2\pi) \approx 5.5$ GHz, represent a two-order-of-magnitude increase in frequency when compared to previous dissipative optomechanics integrated devices[1,32], making our device suitable for applications in the quantum regime such as the optical writing/reading of quantum information into phonons, and optomechanical ground-state cooling. The impact of the mechanically mediated waveguide-cavity interaction was assessed through the acousto-optic transduction in our system, and it displays remarkable signatures on the optomechanical cooling

and heating of the mechanical modes, i.e., dynamical backaction[6]. In fact, we show that dynamical backaction can be either enhanced or suppressed by controlling the interference between dissipative and dispersive contributions. The effects of dissipative scattering were also visualized, for the first time, on the optical spectrum through the phenomenon of optomechanically induced transparency[34,35], adding another tool to control integrated tunable optical delays and classical/quantum memories.

## Results

In Fig. 1b we show the principle of operation of our device. Two optical modes, $\hat{a}_1$ and $\hat{a}_2$, with identical frequency $\omega_0$ and intrinsic loss (absorption and radiation) $\kappa_i$, are mutually coupled with a rate $J$. The system is driven through a waveguide, which carries a coherent field with amplitude $\bar{\alpha}_{in}$ and couples only to resonator 1 yielding an extrinsic loss $\kappa_e$. Furthermore, mode $\hat{a}_1$ is dispersively coupled to a mechanical mode $\hat{x}$ with a frequency pulling parameter $G_{\omega_1}$, inducing a detuning $\Delta\omega(x) = -G_{\omega_1}\hat{x}$ between $\hat{a}_1$ and $\hat{a}_2$. This detuning changes the effective coupling between resonators $\hat{a}_1$ and $\hat{a}_2$, giving rise to a mechanically dependent collective response of the system, described by the supermodes $\hat{a}_+$ and $\hat{a}_-$[36–38].

In general, the supermodes' frequencies ($\omega_\pm$–Fig. 1c) and losses ($\kappa_\pm$–Fig. 1d) will depend on the position of the mechanical oscillator, as their response is a combination of the individual properties of $\hat{a}_1$ and $\hat{a}_2$. In fact, in the absence of the mechanically induced detuning, $\Delta\omega(x) = 0$, the supermodes are simply the differential, $\hat{a}_+ = (\hat{a}_1 - \hat{a}_2)/\sqrt{2}$, and common, $\hat{a}_- = (\hat{a}_1 + \hat{a}_2)/\sqrt{2}$, mode pairs. As such, electromagnetic energy is evenly divided between $\hat{a}_1$ and $\hat{a}_2$. Since the mechanical motion drives the system away from this regime, it leads to an asymmetric optical field distribution among the resonators, and the supermode with larger energy density overlap with cavity 1 will have larger extrinsic losses, due to its coupling to the bus waveguide. This interplay leads to a dissipative coupling[29] ($G_{\kappa_{e_\pm}} = \frac{d\kappa_{e_\pm}}{dx}$) in addition to the usual dispersive coupling $G_{\omega_\pm} = \frac{d\omega_\pm}{dx}$. The joint action

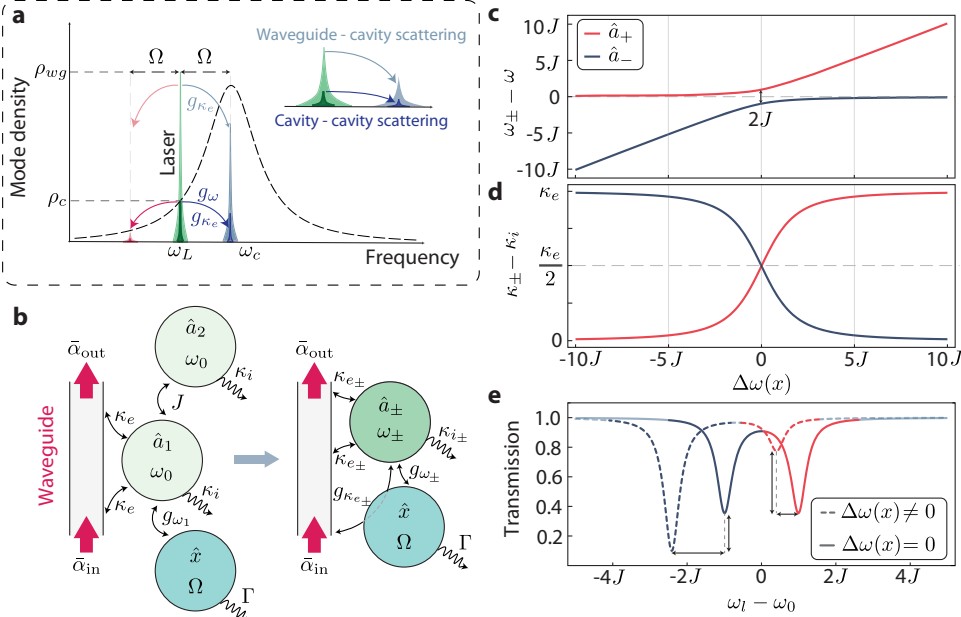

**Fig. 1 | Concept design for dissipative optomechanical coupled cavity.**
**a** Scattering mechanisms in an optomechanical system with both dispersive and dissipative couplings. Photons arriving through a waveguide (mode density $\rho_{wg}$ - light green) can be inelastically scattered into the cavity field depicted by the light blue/red tones. A similar process occurs for cavity pump photons (mode density $\rho_c$ - dark green), generating sidebands in dark blue/red. **b** Diagram for coupled optomechanical cavities with an asymmetric loss induced by the waveguide. The physical response of

the system, depicted on the right, necessarily includes a mechanically dependent extrinsic coupling for both supermodes. **c** Frequency splitting in strongly coupled optical cavities and **d** Supermodes' losses as functions of the frequency detuning between the "bare" resonators, $\Delta\omega(x)$. **e** Transmission spectra for different detuning configurations. The trace colors identify the supermodes as in (**c**) and (**d**). Both their extinction and frequency are sensitive to the mechanically-dependent detuning, as represented by vertical and horizontal double-headed arrows, respectively.

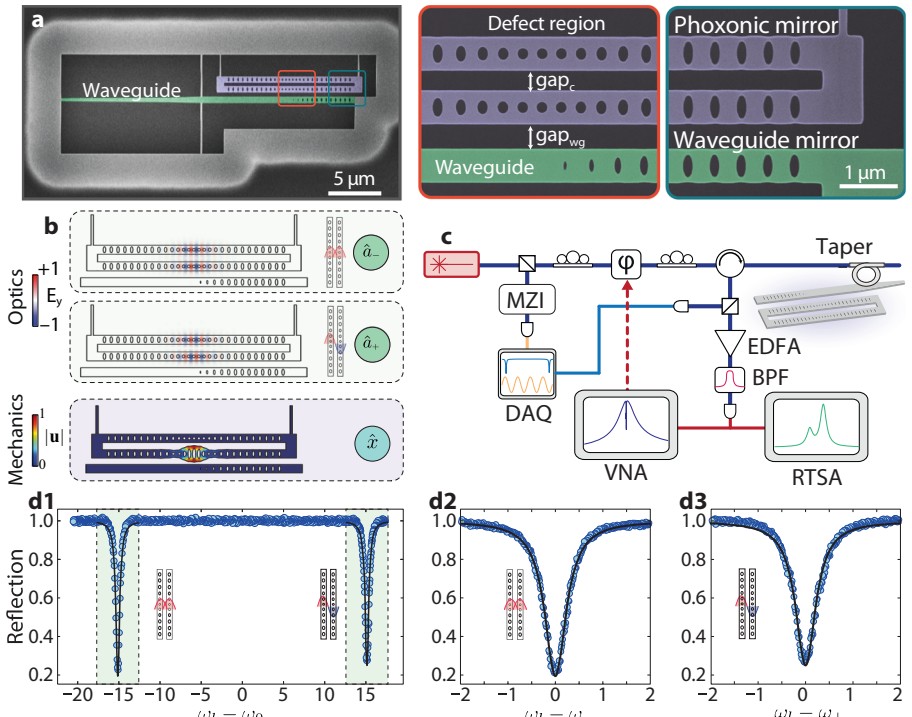

**Fig. 2 | Experimental setup and characterization of coupled optomechanical resonator. a** Scanning electron micrograph of one of the fabricated devices. The image is false-colored to highlight the coupled optomechanical resonators (blue) and tapered waveguide (green). The defect region is responsible for the high-quality confinement of both mechanical and optical modes. The phoxonic (photonic and phononic) mirror suppresses any mechanical coupling between the two nanobeams. **b** Finite-element Method simulations of the optical ($\hat{a}_-$, $\hat{a}_+$) and mechanical breathing ($\hat{x}$) modes of our system. Here, we plot the normalized $y$-component of the electric field, $E_y$, and mechanical displacement $|\vec{u}|$. Although both beams support identical mechanical modes, only one is shown for simplicity. **c** Schematics of the measurement setup. A tunable laser drives our device, which is

accessed using a tapered fiber coupled to the integrated waveguide. The thermo-mechanical noise imprinted in the reflected light is collected and characterized with a real-time spectrum analyzer (RTSA). Coherent spectroscopy is performed using a vector network analyzer (VNA), which modulates the phase of the driving field with an electro-optic phase-modulator ($\phi$). Reflection spectra are measured with a DAQ, along with the output of a Mach-Zehnder interferometer (MZI), which provides the relative frequency of our laser. **d1** Normalized reflection spectrum of the cavity, showing common and differential optical modes. The region around the two resonances is finely scanned with a laser and fitted to a Lorentzian model, as shown in (**d2**) and (**d3**). For the device under analysis, $gap_c = 500$ nm and $gap_{wg} = 450$ nm.

of these effects is illustrated in Fig. 1e, where $\Delta\omega(x)$ leads to variations in both frequencies and extinctions (losses) of the $\hat{a}_\pm$ supermodes.

From our analysis, both the effective dispersive and dissipative couplings are dependent on the individual dispersive coupling of cavity 1, $G_{\omega_1}$. Furthermore, when a weaker intercavity coupling $J$ is present, it results in a steeper avoided crossing between the bare optical modes $a_1$ and $a_2$. This characteristic makes the system more susceptible to mechanical perturbations, thereby increasing the dissipative couplings. However, it is important to ensure that the adiabaticity condition $2J \gg \Omega$ is satisfied (refer to section S1 of the Supplemental Material for a detailed mathematical discussion).

We implement this scheme using a pair of identically designed silicon photonic crystal nanobeams, as shown in Fig. 2a (see Methods). An engineered defect in the central region of both nanobeams supports co-localized optical and mechanical modes with resonances in the optical (1550 nm) and microwave ($\Omega/(2\pi) \approx 5.5$ GHz) bands, respectively. The resonators are separated by a gap, $gap_c$, which evanescently couples their optical modes, in contrast to their acoustic modes which are uncoupled due to an efficient phononic mirror at the clamping edges of the nanobeams. A waveguide is placed laterally to one of the optomechanical cavities and used to probe the device. A photonic mirror defined in the bus waveguide ensures that the output field is efficiently collected. Finite element method simulations for the electromagnetic and acoustic modes of the system are shown in Fig. 2b. The devices were characterized using the setup shown in Fig. 2c, which can probe both optical and mechanical properties of the device at room temperature. The optical response is characterized by

scanning the frequency of a continuous-wave tunable laser and monitoring the cavity's reflection spectrum (see Methods). The coupled optical modes of our system appear as two sharp resonances as shown in Fig. 2d.

The optomechanical interaction imprints information about the thermomechanical motion of the cavity onto the reflected light. Figure 3a shows the optical detector's photocurrent power spectral density, $S_{II}(\Omega)$. Two acoustic modes were found around 5.5 GHz, corresponding to the breathing modes of each silicon nanobeam. Due to natural variations in the fabrication process (see Methods), these modes are non-degenerate and have frequencies $\Omega_1$ and $\Omega_2$. Lorentzian fittings yield mechanical Q-factors $Q_m \approx 2100$ for both modes. Figure 3b shows a density map constructed by stacking multiple photocurrent spectral densities obtained for a range of laser-cavity detunings around the differential optical supermode, $\Delta = \omega_l - \omega_+$; the data in Fig. 3a is represented by the vertical dashed line. The optical mode reflection is also plotted for reference as overlaid points in this map showing a slightly thermo-optic induced bistable regime.

Analyzing $S_{II}(\Omega)$ in Fig. 3b we see clear mechanical signals at $\Delta = \pm\Omega_{1,2}$. Here, input powers are kept low to avoid linewidth modifications in the acoustic spectrum (intracavity photon occupation of $n_c \approx 14$). Considering the response of mechanical modes 1 and 2 as $\Delta$ is varied, we verify a clear imbalance between their signals at $\Delta = \Omega_{1,2}$ and $\Delta = -\Omega_{1,2}$ (see an example of this imbalanced response as a function of the optical detuning in section S6, and Fig. S1 of the Supplemental Material). In the sideband-resolved regime, $\Delta = \pm\Omega_{1,2}$ is the laser-cavity detuning of interest since either the Stokes or Anti-Stokes sidebands

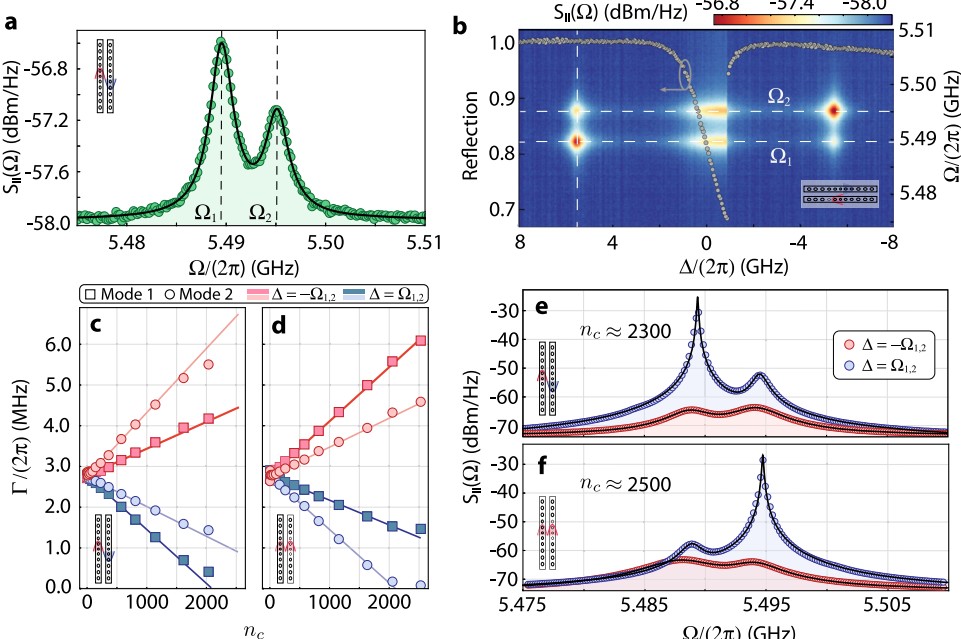

**Fig. 3 | Dissipative and dispersive contributions to optomechanical backaction.** **a** Thermo-mechanical spectrum of the differential optical mode in our device at an input power $P_{in} = 2.7\,\mu W$. Each peak corresponds to the acoustic breathing mode of individual nanobeams. **b** Map of the mechanical spectra as a function of the laser-cavity detuning $\Delta$. The vertical dashed line corresponds to the spectrum in (**a**). The reflection at every $\Delta$ in our measurement is also provided; the gray arrow indicates the associated axis. **c** (**d**) Optically induced modifications to the mechanical linewidth of mechanical modes 1 and 2 for the differential (common) optical modes, as a function of the number of photons in the resonator. **e** Selective phonon lasing contrasted to optomechanical cooling in the differential optical mode. **f** The same analysis for the common optical mode.

generated by the optomechanical interaction are resonantly enhanced, thus leading to appreciable backaction heating or cooling at large input powers. The imbalance in our data suggests that the rates for Stokes and Anti-Stokes scatterings are different for each of the mechanical modes, although with opposing trends for modes 1 and 2 – while at $\Delta = \Omega_1$ the signal for mechanical mode 1 is enhanced, it is suppressed for 2. These results, showing a novel degree of control enabled by the optomechanical interaction, cannot be explained solely within the framework of dispersive optomechanics where the probabilities for Stokes and Anti-Stokes scatterings are identical when considering a system far from ground-state, as is the present case

Our measurements are compatible with a system presenting both dispersive and dissipative optomechanical couplings[1,31]. In this scenario, these two scattering mechanisms can interfere constructively or destructively, depending on the laser-cavity detuning $\Delta$, leading to different scattering rates for anti-Stokes and Stokes bands. Yet, such behavior has not been previously observed in the resolved sideband regime. The root of the observed asymmetry between signals at $\Delta = \pm \Omega_{1,2}$ lies in the different phases of the intracavity and waveguide fields, which enhance dispersive and dissipative scattering mechanisms, respectively. In a frame rotating at the laser frequency, the phase of the cavity field is shifted by $\approx \pi$ between red and blue detunings whereas the waveguide field has a fixed phase. Consequently, the two mechanisms' interference varies from constructive to destructive between blue and red detunings. Furthermore, the opposite behavior of mechanical modes 1 and 2 can be explained through an extension of the setup of Fig. 1b: the mechanically induced detuning $\Delta\omega(x)$ has a different sign if the acoustic mode is coupled to optical mode 2 rather than 1. Since the variation in the optical linewidth is proportional to $\Delta\omega(x)$, mechanical modes 1 and 2 will have dissipative couplings with opposite signs (for a given optical supermode), which in turn results in dissipatively scattered fields with opposite phases.

At sufficiently high optical input powers, driving our system at frequencies resonant with Stokes ($\Delta = \Omega_{1,2}$) or Anti-stokes ($\Delta = -\Omega_{1,2}$)

scattering processes leads to appreciable amplification or cooling of the mechanical modes. We anticipate an imbalance in the efficiency of these processes inherited by the same interference effects between the dispersive and dissipative scattering channels discussed above. In the sideband-resolved limit the extra damping rate of the mechanical mode, $\Gamma^{OM}(\Delta)$, is given by (see S3)

$$\Gamma^{OM}(\pm \Omega) \approx \mp \frac{\left(\mp 2\sqrt{n_c}g_\omega + g_{\kappa_e}\frac{\bar{\alpha}_{in}}{\sqrt{\kappa_e}}\right)^2}{\kappa},\tag{1}$$

where $g_\omega = G_\omega x_{zpf}$ and $g_{\kappa_e} = G_{\kappa_e} x_{zpf}$ are the dispersive and dissipative vacuum optomechanical coupling rates of (any of) the supermodes, and $x_{zpf}$ is the zero-point fluctuation in the mechanical displacement. From Eq. (1), we expect different relative signs of $g_{\kappa_{e_\pm}}/g_{\omega_\pm}$ to benefit opposite backaction effects (cooling or heating). For instance, if $g_\kappa/g_\omega < 0$ the heating process is enhanced, to the detriment of cooling, whereas the converse happens if $g_\kappa/g_\omega > 0$. An interesting feature of Eq. (1) is the dissipative contribution scaling with the normalized input field amplitude $\bar{\alpha}_{in}/\sqrt{\kappa_e}$, which is a factor $\Omega/\kappa_e$ larger than $\sqrt{n_c}$ in the sideband-resolved regime, thus leading to an appreciable dissipative contribution to optomechanical backaction even in the regime of $|g_{\kappa_{e_\pm}}/g_{\omega_\pm}| \ll 1$.

The optical control of the mechanical linewidth in our device is shown in Fig. 3c for the differential optical mode. Due to dissipative coupling and its opposite signs between acoustic modes, one mechanical mode can be preferentially heated or cooled over the other. When driving at the common optical mode the responses of the mechanical modes are swapped, as shown in Fig. 3d. This is once more consistent with the analysis presented in Fig. 1d: the variation in the losses of each of the supermodes $\kappa_\pm$ is the opposite for given a detuning $\Delta\omega(x)$, hence yielding dissipative couplings with different signs for $a_+$ and $a_-$. The solid lines represent fits of the full model of Eq. (1) to the experimental data from which we extract both dispersive

and dissipative coupling rates between all the mechanical and optical modes. The obtained values are displayed in Table 1. For sufficiently large photon occupations $n_c > 2000$ we achieve photon-phonon cooperativities $C = |\Gamma^{OM}|/\Gamma > 1$. This is an important metric for quantum and classical information transfer protocols whose efficiency scales with $C$. Such large cooperativities are evidenced by the mechanical spectra in the differential (Fig. 3e) and common (Fig. 3f) optical modes under blue (red) laser-cavity detunings where clear narrowing (broadening) of the mechanical modes is verified. Interestingly, the presence of the dissipative coupling and its different signals between mechanical modes allows one to selectively induce self-sustained oscillations in each of them.

In the sideband-resolved regime photons and phonons can hybridize and give rise to photon-phonon polaritons[27,34,35], marking the onset of the strong-coupling regime of optomechanics. This phenomenon is more noticeable if the system is driven resonantly with the mechanical mode, i.e. $\Delta = \pm \Omega$. In this case, transmission and reflection measurements clearly display features arising from the mechanical lineshape, appearing as either dips or peaks in the optical spectrum and leads to substantial modifications in the group velocity of light passing through the device. This effect is the optomechanical analog of electromagnetically induced transparency/absorption in atomic[39] and solid-state systems[40] and hence named optomechanically induced transparency or absorption (OMIT/OMIA).

### Table 1 | Dissipative and dispersive coupling rates for each pair of optical and mechanical modes extracted from fits of the full model leading to Eq. (1)

|  | Mech. Mode 1 | | Mech. Mode 2 | |
|---|---|---|---|---|
|  | $g_\omega/2\pi$(kHz) | $g_\kappa/2\pi$(kHz) | $g_\omega/2\pi$(kHz) | $g_\kappa/2\pi$(kHz) |
| $a_+$ | $352 \pm 10$ | $-2.8 \pm 0.1$ | $378 \pm 10$ | $3.4 \pm 0.1$ |
| $a_-$ | $356 \pm 10$ | $3.8 \pm 0.1$ | $367 \pm 10$ | $-3.1 \pm 0.1$ |

Regardless of the recent progress in dissipative optomechanical systems, the observation OMIT/OMIA remains elusive since the resolved sideband regime has not yet been reached[1,31,32]. Our device allies large mechanical frequencies and dissipative couplings, making it uniquely suited for such demonstration. OMIT is achieved by setting a carrier laser, at frequency $\omega_l$, such that its mechanically scattered Anti-Stokes sideband is resonant with an optical mode, i.e., $\Delta = -\Omega$. Fundamentally, the optomechanical interaction generated by the pump causes transitions annihilating one phonon ($n_m \to n_m - 1$), where $n_m$ denotes the phonon occupation. On the optical domain, energy conservation requires a frequency-shifted photon (population $n_p$) to be created ($n_p \to n_p + 1$). A probe beam, at frequency $\omega_l + \Omega_{mod}$, induces phonon-number conserving transitions. When $\Omega_{mod}$ approaches $\Omega$, interference between mechanically scattered photons and the probe beam results in a transparency window in the probe's reflection spectrum. In our system, the depth and width of the dip in the optical spectrum are largely affected by both dispersive and dissipative mechanical scattering mechanisms, leading to the first observation of dissipative signatures in OMIT/OMIA. A generalized scheme for OMIT, accounting for dissipative and dispersive effects, is summarized in Fig. 4a.

We assess the changes in the reflected optical spectra by phase-modulating our strong input laser (pump) and generating a weak probe beam. The modulation frequency, $\Omega_{mod}$, is varied using a vector network analyzer (VNA), which also measures the beating signal between the pump and the probe beams, yielding the scattering parameter $S_{21}(\Omega_{mod})$ (see Methods). A typical curve for $|S_{21}(\Omega_{mod})|$ is shown in Fig. 4b, where we highlight the aforementioned transparency window, at a frequency matching the nanobeams' breathing modes.

In Fig. 4c we zoom into the transparency windows measured for both optical modes for a range of input powers and detunings $\Delta \approx -\Omega_{1,2}$. Remarkably, we selectively induce deeper and wider dips at $\Omega_1$ ($\Omega_2$) by driving the common (differential) optical mode. We repeat this experiment for a range of input powers and detunings $\Delta \approx -\Omega_{1,2}$. A

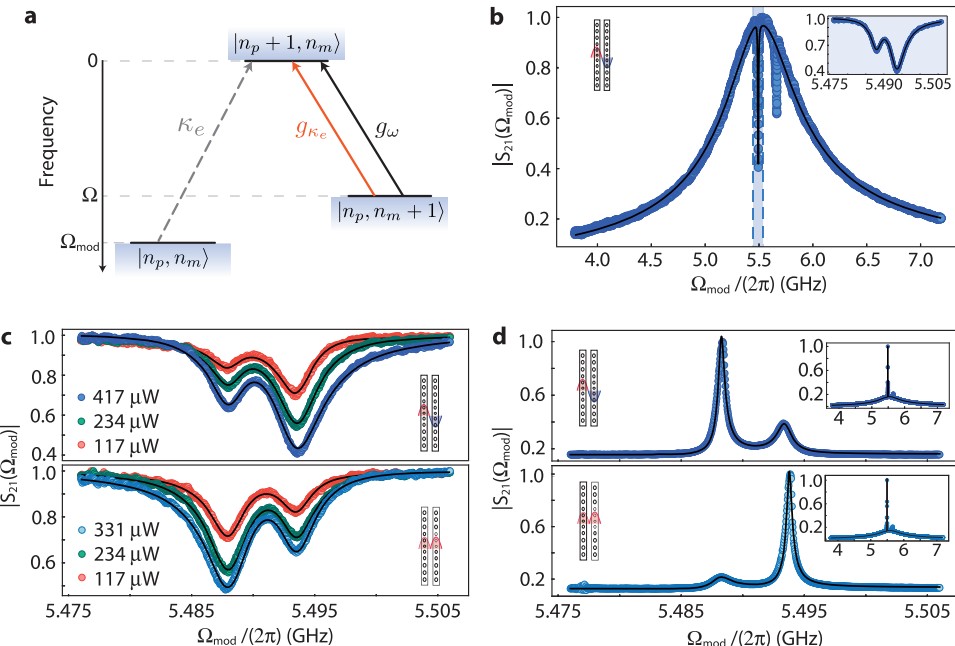

**Fig. 4 | Optomechanical induced transparency from dissipative and dispersive coupling. a** Level diagram of OMIT. The probe beam induces phonon-conserving transitions while the mechanically scattered photons necessarily decrease the phonon population. The two optomechanical coupling mechanisms (dissipative and dispersive) interfere to generate the transparency window. **b** Scattering parameter magnitude, $|S_{21}|$, as a function of the phase-modulation frequency, $\Omega_{mod}$.

Data is shown for the differential optical mode. Inset: Transparency window as highlighted in blue. **c** Transparency windows for the differential (top) and common (bottom) optical modes as a function of input power. **d** Absorption windows for the differential (top) and common (bottom) optical modes. Input powers are indicated through the matching colors with (**c**).

model for $|S_{21}(\Omega_{mod})|$ including optomechanical effects is fit to the data and yields $g_{\kappa_e}$ and $g_\omega$ consistent with our previous characterization (see Methods). Our experiments are limited by the thermo-refractive response of the cavity, which inhibits access to $\Delta = -\Omega_{1,2}$ at large input powers. This could be mitigated by cooling the present device to temperatures $T < 20$ K, where the mechanical Q-factors are increased and optically induced heating is reduced. In Fig. 4d we display complementary OMIA measurements, where the pump is set to $\Delta \approx \Omega_{1,2}$, and once more the mechanical mode selectivity is observed. Our theoretical model, fed with the parameters extracted from the OMIT experiments, was able to reproduce the OMIA results with great accuracy, further validating our analysis.

We demonstrated the first dissipative optomechanical system operating in the good-cavity regime. To the best of our knowledge, our results represent a two-order of-magnitude leap in the acoustic frequencies (5.5 GHz)[32] with a tenfold increase in dissipative couplings ($|G_{\kappa_{e_\pm}}| \approx 1$ GHz/nm)[31] when compared to previous literature. This allowed the first observations of dissipative optomechanics signatures on both optical and mechanical spectra, despite the modest figure of $|g_{\kappa_e}/g_\omega| \approx 1\%$. Our data and theoretical modeling showed that the waveguide field enhancement in the dissipative optomechanical interaction (in contrast to the intracavity field enhancement in dispersive optomechanics) boosts the dissipative contribution in the interplay between acoustic and optical responses in our device, mitigating the shortcomings of small $g_{\kappa_e}$. In our system, an approximate sixfold enhancement in $g_{\kappa_e}$ could be achieved solely by reducing the photon tunneling rate, enabling, for example, an almost complete cancellation of either the backaction heating or cooling in a given optical/mechanical mode pair. In this condition, one obtains almost independent optomechanical control of the virtually degenerate mechanical modes 1 and 2 (see section S7, and Fig. S2 of the Supplemental Material). Individual addressing of acoustic modes could also be accomplished in dual-tone experiments with simultaneous red- and blue-detuned lasers. By tuning the input power of each tone one could completely cancel backaction on a given mechanical mode. In the classical realm, this prospect is potentially interesting for sensing applications, as in the generation of mechanical exceptional points[41], where tuning losses of different mechanical modes is necessary. In the quantum regime, artificial atoms such as microwave qubits and color centers could be individually addressed by each mechanical mode, creating a scalable tool for the accurate control of quantum information[42,43], moreover, this setup could be further explored for quantum sensing, e.g., in backaction-evading experiments[22].

A critical analysis of Eq. (1) reveals that dissipative scattering could overcome its dispersive counterpart if $g_\omega < g_{\kappa_e}\Omega/(2\kappa_e)$. State-of-the-art dispersive systems currently operate at $g_\omega/(2\pi) \approx 1$ MHz, thus, the predominance of the dissipative coupling requires $g_{\kappa_e} > 54$ kHz (for the mechanical frequencies and extrinsic losses of our device), which is over one order of magnitude larger than the values reported here. However, as shown in our data, dispersive and dissipative contributions can constructively interfere and therefore having nonzero $g_{\kappa_e}$ can be advantageous even at smaller values. Complementarily, a plethora of design routes[44–46] (with integrated photonic analogs) and toolboxes[28] for dissipative optomechanics can be used to further improve our figures. Advances in this field could also be obtained by exploring more complex systems such as cavity exciton-polaritons[47], and electromechanical resonators[25].

As a last remark, the waveguide-cavity scattering process discussed and demonstrated in this work remains vastly unexplored. It enables the optomechanical interaction to take place even in the absence of circulating pump photons in the cavity, which is potentially an advantage over the dispersive process. One could harness this advantage by introducing devices that entirely suppress the intracavity field, as in a system interacting with two driving channels destructively interfering at the cavity. In this case, the dissipative coupling allied to

strong excitation fields could still generate mechanically scattered photons in the optical mode, while the resonator's optical response would be cloaked against unwanted linear/nonlinear absorption arising from the buildup of pump photons in the cavity mode. This is currently one of our limitations in achieving high optomechanical cooperativities.

## Methods

### Fabrication
The device manufacturing procedure follows a basic CMOS-compatible top-down approach following the recipe developed for Ref. 24. Electroresist polymer (CSAR-09) is spun at 2000 rpm for 1 min on top of an SOI (silicon on insulator) wafer die. The device designs are patterned on the electroresist using a 100 kV e-beam lithography tool followed by a development step immersing the chip for 1 minute in the pentyl-acetate solution. The device is then transferred to the silicon layer using a $SF_6+O_2$ plasma etch at cryogenic temperatures. The residual electroresist is removed using a piranha process ($H_2SO_4$:$H_2O_2$ - 3:1) followed by silica etch to release the structure using HF (hydrofluoric acid) solution for 3 min.

We attribute to fabrication variations the observed shift in the mechanical frequencies of resonators 1 and 2 to mainly: (a) global electron dosing variations due to the writing pattern, (b) turbulence during the insertion of the chip into the chemical solution for electron resist development, which can produce variations in the development rate, and (c) temperature gradient in the chip during plasma etching influencing the local etch rate. Further details on the reproducibility of our devices and variations in their critical properties can be found in the Supplementary Material of Ref. 24.

### Optical characterization
The optical characterization was performed using a Toptica CTL 1550 laser. From the data shown in Fig. 2 we infer a photon tunneling rate $J/2\pi \approx 15$ GHz from the mode-splitting. A Lorentzian model fit to each transmission dip extracts their linewidths, $(\kappa_-, \kappa_+)/2\pi = (546, 515)$ MHz, and extrinsic losses $(\kappa_{e_-}, \kappa_{e_+})/2\pi = (153, 129)$ MHz.

### Optomechanical transduction
The reflected light was captured using a fast photodetector (Discovery Semiconductors DSC30S) and a real-time spectrum analyzer (Agilent PXA N9030A). The laser frequency is swept from red to blue detunings, allowing the access of $S_{II}(\Omega)$ at $\Delta = \pm \Omega_1$, which becomes challenging for high input powers due to optical nonlinearities leading to a bistable behavior on the optical spectrum[48,49]. In our experiment, optical bistability is observed for optical powers as low as $1\,\mu W$. Transduction with optical power below the bistable regime is shown in S6.

The results are modeled using input-output theory to describe the photocurrent power spectral density $S_{II}(\Omega)$ shown in Fig. 3. The classical optical field amplitude, $a(t)$, is given by

$$\dot{a}(t) = i(\Delta + G_\omega x)a(t) - \frac{\kappa + G_{\kappa_e}x}{2}a(t) - \sqrt{\kappa_e}\bar{\alpha}_{in} - \frac{G_{\kappa_e}x}{2\sqrt{\kappa_e}}\bar{\alpha}_{in}. \quad (2)$$

Linearizing this equation around stationary coherent amplitudes $\bar{x}$ and $\bar{a}$, i.e., $a(t) \rightarrow \bar{a} + \delta a(t), x(t) \rightarrow \bar{x} + \delta x(t)$, and keeping terms only to first order in the fluctuations $\delta x(t)$, $\delta a(t)$, we arrive at the output field amplitude

$$\alpha_{out}(t) = \bar{\alpha}_{in} + \sqrt{\kappa_e}(\bar{a} + \delta a(t)) + \frac{G_{\kappa_e}\bar{a}}{2\sqrt{\kappa_e}}\delta x(t). \quad (3)$$

The photocurrent $I(t)$ is proportional to $|\alpha_{out}(t)|^2$, whose fluctuations are given by $\delta I(t) = \bar{\alpha}_{out}\delta\alpha^*_{out}(t) + \bar{\alpha}^*_{out}\delta\alpha_{out}(t)$. Using a spectrum analyzer, one measures the power spectral density of $\delta I(t)$,

**Table 2 | Dissipative and dispersive coupling rates for each pair of optical and mechanical modes extracted from OMIT measurements**

| | Mech. Mode 1 | | Mech. Mode 2 | |
|---|---|---|---|---|
| | $g_\omega/2\pi$(kHz) | $g_\kappa/2\pi$(kHz) | $g_\omega/2\pi$(kHz) | $g_\kappa/2\pi$(kHz) |
| $a_+$ | $349 \pm 8$ | $-2.8 \pm 0.1$ | $405 \pm 10$ | $3.6 \pm 0.1$ |
| $a_-$ | $373 \pm 10$ | $4.0 \pm 0.1$ | $372 \pm 10$ | $-3.1 \pm 0.1$ |

$S_{II}(\omega) = \int d\tau e^{i\omega\tau} \langle \delta I(\tau)\delta I(0) \rangle$. Finding this quantity requires moving into a frequency domain description in Eq. (3), where the optical field $\delta a(\omega)$ is written in terms of $\delta x(\omega)$. Finally, $S_{II}(\omega)$ can be written in terms of the mechanical power spectral density, $S_{xx}(\omega)$, which is independent of the detuning for low input powers. The transduction function between $S_{xx}(\omega)$ and $S_{II}(\omega)$ is given in S2 and depends on the detuning $\Delta$, $\omega$ and other properties such as the extrinsic coupling and total losses of the optical mode under analysis.

## Optomechanically induced transparency

The phase-modulation on the pump frequency, $\Omega_{\text{mod}}$, is varied using a vector network analyzer (Agilent PNA E8362C). The carrier and probe are reflected from the cavity and interfere at a fast photodiode, yielding a fluctuating photocurrent that carries the information of any optomechanical contribution to the probe's spectrum. This photocurrent is fed to the VNA which measures the scattering parameter $S_{21}(\Omega_{\text{mod}})$.

The input laser phase modulation is described as $\bar{\alpha}_{\text{in}} \to \bar{\alpha}_{\text{in}} e^{-i\phi_0 \sin(\Omega_{mod}t)}$ in Eq. (2). For weak modulations, $\phi_0 \ll 1$, the system is effectively driven by a strong pump tone, at frequency $\omega_l$ and two probes at $\omega_l \pm \Omega_{\text{mod}}$. In the sideband-resolved regime, $\Omega_{\text{mod}} \gg \kappa$ and the cavity response filters out one of the probe tones. The remaining sideband induces a cavity amplitude $a_+$, which is given by

$$a_+ \approx -\frac{\sqrt{\kappa_e}\bar{\alpha}_{\text{in}}\phi_0}{-2i\left(\Delta + \Omega_{\text{mod}}\right) + \kappa + \frac{n_c(\Delta g_{\kappa_e} - 2\kappa_e g_\omega)^2}{\kappa_e^2(\Gamma - 2i\Omega_{mod} + 2i\Omega)}}, \tag{4}$$

where we assumed $\Delta < 0$ and $|\Delta| \gg \kappa$, leading to the OMIT configuration. Here $\Omega$ is the mechanical frequency of the acoustic mode under analysis and $\Delta$, $\kappa_e$, and $\kappa$ already include static shifts due to an average mechanical displacement $\bar{x}$.

We immediately verify that the optical susceptibility is strongly dressed by the optomechanical interaction when $\Omega_{\text{mod}} \approx \Omega$. This information is naturally imprinted in the reflection spectrum of the probe, which is directly connected to the magnitude of the scattering parameter $S_{21}(\Omega_{\text{mod}})$, measured in our experiment. A full derivation of this equation is presented in S4.

An extension of Eq. (4) can be used to derive a model for $|S_{21}(\Omega_{\text{mod}})|$ (see S4). Fixing the ratio $g_{\kappa_e}/g_\omega$ using the results from Fig. 3 and fitting $|S_{21}(\Omega_{\text{mod}})|$ to the data in Fig. 4 we obtain the results in Table 2.

## Data availability

Experimental data and script files required for generating each figure can be found in the ZENODO repository, accessible via the following link: https://doi.org/10.5281/zenodo.8072538[50].

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

## Acknowledgements
This work was supported by São Paulo Research Foundation (FAPESP) through grants 19/09738-9, 20/15786-3, 19/01402-1, 18/15577-5, 18/15580-6, 18/25339-4, 22/07719-0, Coordenação de Aperfeiçoamento de Pessoal de Nível Superior—Brasil (CAPES) (Finance Code 001), the European Research Council (ERC CoG Q-ECHOS, 101001005), and by the Netherlands Organization for Scientific Research (NWO/OCW), as part of the Frontiers of Nanoscience program, as well as through Vrij Programma (680-92-18-04). We thank Prof. Fanny Béron and Prof. Kleber Pirota for providing access to the VNA used for OMIT/OMIA experiments and Dr. Felipe Santos for helpful comments.

## Author contributions
A.G.P, P.V.P. and T.P.M.A. devised and planned the experiment; R.B. fabricated the samples with design help from S.G. and T.P.M.A; A.G.P, P.V.P. performed measurements, data analysis and FEM simulations with contributions from G.S.W and T.P.M.A; S.G., G.S.W. and T.P.M.A. supervised the project. All authors contributed to the discussions and preparation of the manuscript.

## Competing interests
The authors declare no competing interests.
