## [Peer Review File · Nature Communications]

Dissipative Optomechanics in High-Frequency Nanomechanical ResonatorsREVIEWER COMMENTS

Reviewer #1 (Remarks to the Author):

Referee's report on "Waveguide-Cavity Scattering in High-Frequency Dissipative Optomechanics" by A G Primo et al.

1. The research reported is very timely and of high quality.
2. The main result of the manuscript is the demonstration of an optomechanical structure where the dominant mechanism of transduction is in the dissipative regime at room temperature, in a relatively high Q cavity realised in a broadly used material with up-scalable associated technology. The second results worth mentioning is the combined two-orders of magnitude increase in the mechanical frequency reaching 5.5 GHz and the 10x improvement in dissipative coupling now at approximately 1 GHz/nm.
3. The manuscript will be of significance in the field of optomechanical systems, in which the community strives towards its use in quantum technologies, specifically quantum computation. It will most likely resonate with the communities working on non-linear optical phenomena in both strong and weak coupling regimes.
4. The results reported constitute a sound step forward pushing further the experimental state of the art, specifically in silicon-based optomechanics. It builds on the body of knowledge of the community. The only proviso is that the presentation of the main manuscript is far too dense. While it shows that the authors command the concepts, the experimental methods and are aware of the literature, it is rather heavy to read. The reading and comprehension of the Supplementary Information is a must, if the results are to be appreciated for their significance. The style certainly does not do much to open up the field to a wider community and remain firmly anchored in the highly specialised interface between optomechanics and non-linear optics. This is particularly relevant since the authors suggest possible developments toward the use of dissipative optomechanics in quantum information processing.
5. The conclusions are well supported by both the experimental work and the theoretical/simulations parts.
6. The methodology is sound but there are details missing if the experiment were to be reproduced, especially details of the physical realisation: the optomechanical structure.:
 - a. Mention is made of the "fabrication imperfections (lines 183-184) however, no details are given of the extent and kind of these.
 - b. Within what sigma do the critical dimensions vary?
 - c. How smooth is smooth (line 449). Any way to quantify either in nm or in terms of the dominant critical dimension?
 - d. What are the dominant non-deterministic imperfections?
 - e. Are the two nanobeam only supported by the two wires attached to the main frame of the suspended structure on one side, as shown in figure 2a? And, if so, what role do these have in offering a stable system without additional torsional/vibrational modes? Perhaps such modes do appear in a different part of the spectrum. Could the authors confirm?
7. It would be very helpful to have a clear statement of what is strong coupling in terms of the laser pump and the optical beam modulation. There seems to be a degree of ambiguity in the description of the simulation work, as presented in the Supplementary Information, where approximations are made suitable for the weak coupling regime (SI. line S91).
8. Thus, perhaps the only recommendation to the authors in addition to the details needed to reproduce the experiment, is to enhance the clarity. Towards (a) enhancing accessibility for other proposals on quantum information processing and (b) allowing the analysis to be applied to other configurations.

Reviewer #2 (Remarks to the Author):

The paper describes experimental results obtained with an optomechanical system composed of two optical modes, in which only one of the two is driven and coupled to an optical waveguide. Each optical mode is coupled to a high frequency co-localized mechanical mode. The novel feature of the system is that it is operated in a regime where the dissipative optomechanical coupling becomes appreciable and comparable to the more standard dispersive one. The main result of the paper is the detection of experimental signatures of dissipative coupling optomechanics in the resolved sideband regime, as also stressed by the authors. In fact previous demonstrations of dissipatively coupled optomechanical devices were carried out in the bad cavity regime, which hinders many interesting interference phenomena.

The paper is clearly written, and together with the supplementary material, it provides a quite clear description of the physics. The interference between the dispersive and the dissipative coupling is responsible for interesting constructive and destructive phenomena visible in the transmission, in the modification of the mechanical behavior, and in the OMIT and OMIA spectra.

However, I am not convinced that the present paper is relevant enough for the broad scientific audience of Nature Communications. In fact, the results, even though interesting, are quite technical, and they seem to me relevant only for the community working on optics and optomechanics. Moreover, I am not convinced that the dissipative coupling can be controlled and improved so that it may become a useful tool for approaching the quantum regime of optomechanics, which has been already achieved with various devices using dispersive coupling. I am not sure that the relevant parameters, especially the coupling with the waveguide κ_e and the tunneling J , can be tuned so that one can have strong dissipative coupling optomechanics.

In this respect the final sentence of the abstract "The present demonstration opens a path to strongly dissipative optomechanical devices with nearly noiseless operation in the quantum regime." seems to me too strong and not justified by the results of the paper.

A minor point is that the author could mention as possible realizations of dissipative optomechanical coupling also hybrid atom-optomechanical systems. In fact the optical mode uncoupled to the waveguide could be replaced by an atomic system coupled only to the cavity mode. One could have similar effective dissipative coupling and interference phenomena (see for example C. Genes et al., Phys. Rev. A 80, 061803(R) (2009) and references therein.

In conclusion I suggest publication of the present paper in more specialized journal.

Reviewer #3 (Remarks to the Author):

The manuscript by Primo et al. reports an optomechanical system with two coupled photonic crystal cavity modes, of which only one is coupled to an input-output channel. As a result, the displacement of a mechanical mode changes the coupling rate of the optical eigenmodes to the output channel, giving an effectively dissipative optomechanical coupling. This effect is observed in the context of transduction of thermomechanical motion, backaction, and optomechanically-induced transparency. While dissipative optomechanical coupling has been observed in other systems, the authors report in this work the first realization in a device in the resolved sideband regime.

The experimental system is state-of-the-art, the quality of the observed data is high, and the observations are clearly described by the theoretical model that includes dissipative coupling that is given in the paper. Dissipative coupling has been considered since well over a decade, but did not find wide adoption in cavity optomechanical applications. This work certainly presents an interesting experimental advance, bringing dissipative coupling to nanoscale, sideband-resolved systems. As a main motivation for the work, the authors argue that dissipative coupling could help perform quantum optomechanical experiments, in particular coherent transduction, with fewer photons resonating in the optical cavity. As in current experiments absorption from intracavity pump photons is a main limitation, this makes any significant step towards a solution to this problem one with high impact.

However, specifically on this last point I have some doubts that are not resolved in the current manuscript. I would therefore urge the authors to change the manuscript in order to address the following:

The idea that the system described in their work allows optomechanical effects of a given strength with fewer intercavity photons than for standard dispersive coupling is posed without theoretical proof or other underpinning. The idea is sketched conceptually in Fig. 1a. The accompanying text in the introduction speaks of a 'direct scattering' to the waveguide that could significantly reduce the required intracavity pump power. However, the work does not demonstrate for what parameters this advantage will become apparent. This is especially important since the dissipative coupling in this coupled-cavity system is fundamentally originating from a dispersive optomechanical coupling in one of the individual cavities. It is therefore not obvious that linear optomechanical effects and in particular coherent state transduction could be enhanced at reduced intracavity photon number.

Given that the authors provide a theoretical framework, it should be straightforward to provide a calculation that supports the above claim, which forms the main motivation of the work. For instance, the authors could calculate the optomechanical cooling rate (as an example optomechanical effect) versus the intracavity photon number, for two different cases: an individual cavity with dispersive coupling G_{ω_1} and intrinsic loss rate κ_1 coupled to a waveguide at rate κ_e , and that same system optically coupled to another cavity at rate J with the same intrinsic loss rate. (Alternatively, one could also consider the depth of an OMIT dip instead of the cooling rate, for example.)

I believe the above would yield a fair comparison, in line with the studied system, that would provide a strong motivation for the presented work. I note that it would not be sufficient to instead point at the difference between the case with combined effective dispersive and dissipative coupling and the case with only the effective dispersive coupling, as observed in Figs. 4 and 5 and denoted in eq. 1: In the coupled-cavity system, both the effective dispersive and dissipative coupling to the optical supermodes are altered with respect to their values in the single-cavity system. They are thus related, and cannot be made arbitrarily large or small compared to the properties of the individual cavities in the system. In summary, one should not compare effective dispersive and dissipative coupling in a coupled-cavity system, but rather compare the dissipative coupling that emerges in the coupled system to the dispersive coupling that can be achieved with a single cavity. That is, after all, how one would choose between one or the other in an application.

If the result of the suggested comparison would show a clear and strong benefit for the coupled-cavity case, I think that could contribute to the clarity of the motivation, as well as support the discussion paragraph. If it is not unambiguous, I propose that the authors revise the claims that dissipative coupling, in this coupled-cavity system, will surely lead to a reduced power requirement for optomechanical applications such as transduction in both the abstract and the introduction.

Finally, I note that the authors stress the importance of achieving dissipative coupling in the resolved sideband regime. It would be helpful to the broader audience if they would discuss this need in more depth, specifically in the context of dissipative coupling. It should be noted that many theory works on dissipative coupling (e.g. ref. (26) by Elste et al. and ref. (36) by Yanay et al.) in fact stressed the ability of this mechanism to yield ground state cooling and quantum control outside the resolved sideband regime as a defining advantage, setting it apart from

dispersive coupling. I suggest that the authors explain what is known in the field, i.e. that while dissipative coupling can lead to quantum control outside the resolved sideband regime (mechanical frequency smaller than total optical decay rate κ), the ratio of mechanical frequency to intrinsic optical dissipation rates should still be large. That is what is demonstrated in this work.

Response Letter: Waveguide-Cavity Scattering in High-Frequency Dissipative Optomechanics

André G. Primo, Pedro V. Pinho, Rodrigo Benevides, Simon Gröblacher, Gustavo S. Wiederhecker, and Thiago P. Mayer Alegre.

Summary of response

We express our gratitude to the referees for their careful review of our manuscript. Their comments and suggestions were thoroughly incorporated into the revised version, significantly enhancing the quality and clarity of the work. In particular, we have made the following notable improvements to the manuscript:

1. We have significantly remodeled the text towards a clearer and simpler view of the phenomena depicted by our experiment.
2. We have added to the text a more thorough description of the interference between dispersive and dissipative couplings for both optical and mechanical modes. Our new text points to an interesting route to the individual control of different mechanical modes in a single device.
3. Fig. 4 was updated and now displays data for the responses of both mechanical and optical modes.
4. We have revised the claims surrounding a clear advantage of the dissipative optomechanical approach for quantum transduction experiments and have expanded the prospects of our work, providing more details on how to improve the next generation of experiments.
5. We have significantly expanded the Methods section where technical details of our experiment can now be found.
6. Data supporting our findings and scripts for generating all figures are now available using Zenodo repository at <https://doi.org/10.5281/zenodo.8072538>.

A marked-up version has been provided for easy comparison. We are confident that the revisions have significantly improved the manuscript, and made more clear the impact of our work for the broader community. We appreciate the referees for their invaluable input.

Reviewer #1 (Remarks to the Author):

Referee's report on "Waveguide-Cavity Scattering in High-Frequency Dissipative Optomechanics" by A G Primo et al.

1. The research reported is very timely and of high quality.
2. The main result of the manuscript is the demonstration of an optomechanical structure where the dominant mechanism of transduction is in the dissipative regime at room temperature, in a relatively high Q cavity realised in a broadly used material with up-scalable associated technology. The second results worth mentioning is the combined two-orders of magnitude increase in the mechanical frequency reaching 5.5 GHz and the 10x improvement in dissipative coupling now at approximately 1 GHz/nm.
3. The manuscript will be of significance in the field of optomechanical systems, in which the community strives towards its use in quantum technologies, specifically quantum computation. It will most likely resonate with the communities working on non-linear optical phenomena in both strong and weak coupling regimes.

We are grateful for the extremely positive feedback regarding the quality and significance of our manuscript.

4. The results reported constitute a sound step forward pushing further the experimental state of the art, specifically in silicon-based optomechanics. It builds on the body of knowledge of the community. The only proviso is that the presentation of the main manuscript is far too dense. While it shows that the authors command the concepts, the experimental methods and are aware of the literature, it is rather heavy to read. The reading and comprehension of the Supplementary Information is a must, if the results are to be appreciated for their significance. The style certainly does not do much to open up the field to a wider community and remain firmly anchored in the highly specialised interface between optomechanics and non-linear optics. This is particularly relevant since the authors suggest possible developments toward the use of dissipative optomechanics in quantum information processing.

We agree with the referee's observation that the initial presentation of our work and findings was excessively dense. Accordingly, we revised the text, incorporated qualitative arguments, and included a more accessible discussion for the broader scientific community. In particular, we enriched the final remarks section with a discussion on the field of quantum information processing, which we believe will be of significant interest to researchers in this area.

Action: The manuscript was revised and offers a simpler and more intuitive view of the phenomena.

5. The conclusions are well supported by both the experimental work and the theoretical/simulations parts.

We thank the referee for his/her kindfull comment.

6. The methodology is sound but there are details missing if the experiment were to be reproduced, especially details of the physical realisation: the optomechanical structure.:
- Mention is made of the “fabrication imperfections (lines 183-184) however, no details are given of the extent and kind of these.

In the original text “fabrication imperfections” encompassed all uncertainties present in the microfabrication stages required for devices such as photonic crystal cavities. However, we acknowledge that the use of this term is ambiguous and thus have changed to “fabrication variations” and provided further explanation in their roots as seen in methods.

Action: Included a discussion in “Methods” regarding the roots of fabrication fluctuations in our system.

- Within what sigma do the critical dimensions vary?

The devices in this work are fabricated according to the previously established recipe reported in Ref. 24 of the main text. In particular, in its Supplemental Material the variance in the optical and mechanical frequencies of these resonators are shown for a large ensemble of devices (see figure below).

FIG. S1. a) Histogram of the wavelength of the optical resonances from the two chips used in this experiment. The standard deviation of the distributions are 1.9 nm and 1.7 nm and the distance between the centers of the distributions is 0.5 nm. b) Histogram of the mechanical resonance frequency. In this case, the standard deviation of the distributions are 12 MHz and 14 MHz, respectively, with a distance between the centers of the distributions of 5 MHz. In both figures the chip with device A is in blue, the chip with device B is in orange and they have a total of 121 and 162 working devices respectively.

For the purpose of this work, we solely aimed at obtaining a high degree of similarity between the two nanobeams, and because of the limitations pointed in c), we cannot provide the referee an accurate estimate of the variations on internal dimensions.

- c. How smooth is smooth (line 449). Any way to quantify either in nm or in terms of the dominant critical dimension?

It is possible to obtain an estimate of the roughness at the top facet of the cavity by performing an AFM of the sample surface, which is likely on a sub-nanometer scale given the high optical quality factor found for the devices used in the paper. While we acknowledge that measuring roughness in the internal regions would provide a more complete understanding of the device's optical and mechanical properties, conventional techniques such as profilometry or AFM are limited to the surface and cannot access internal regions.

- d. What are the dominant non-deterministic imperfections?

The critical dimensions of the system, namely, the holes composing the phoxonic mirror, are most susceptible to fabrication fluctuations due to:

1. Global electron dosing variations due to the writing pattern.
2. Turbulence induced during the insertion of the chip into the chemical solution for electron resist revelation, which can produce fluctuations in the revelation rate.
3. Temperature gradient in the chip during plasma etching influencing the local etch rate.

Action: Included a discussion in “Methods” regarding the roots of fabrication imperfections in our system.

- e. Are the two nanobeam only supported by the two wires attached to the main frame of the suspended structure on one side, as shown in figure 2a? And, if so, what role do these have in offering a stable system without additional torsional/vibrational modes? Perhaps such modes do appear in a different part of the spectrum. Could the authors confirm?

We thank the referee for his/her careful comment. It is true that the only structure supporting our cavities are the silicon tethers in Fig. 2a. Nevertheless, we do not observe any torsional modes near the 5.5 GHz range, albeit they should appear in the mechanical spectrum, at frequencies far below the GHz range and with smaller optomechanical couplings due to their increased motional masses and reduced overlap with the optical mode.

As we aimed at demonstrating high optomechanical couplings at high frequencies, we did not investigate such torsional modes.

7. It would be very helpful to have a clear statement of what is strong coupling in terms of the laser pump and the optical beam modulation. There seems to be a degree of ambiguity in the description of the simulation work, as presented in the Supplementary Information, where approximations are made suitable for the weak coupling regime (SI. line S91).

We thank the reviewer for his comments. We included a mathematical statement of what “weak coupling” means for our devices in the Supplemental Material, i.e. that the cooperativity $C \ll \kappa/\Gamma$, meaning that the system is far from the mode splitting regime.

Action: Included an explicit statement of what are the underlying hypotheses while working in the “weak-coupling regime”.

8. Thus, perhaps the only recommendation to the authors in addition to the details needed to reproduce the experiment, is to enhance the clarity. Towards (a) enhancing accessibility for other proposals on quantum information processing and (b) allowing the analysis to be applied to other configurations.

We appreciate the referee’s comments and suggestions, which have played a crucial role in improving the quality and clarity of our manuscript.

Reviewer #2 (Remarks to the Author):

- The paper describes experimental results obtained with an optomechanical system composed of two optical modes, in which only one of the two is driven and coupled to an optical waveguide. Each optical mode is coupled to a high frequency co-localized mechanical mode. The novel feature of the system is that it is operated in a regime where the dissipative optomechanical coupling becomes appreciable and comparable to the more standard dispersive one. The main result of the paper is the detection of experimental signatures of dissipative coupling optomechanics in the resolved sideband regime, as also stressed by the authors. In fact previous demonstrations of dissipatively coupled optomechanical devices were carried out in the bad cavity regime, which hinders many interesting interference phenomena.

The paper is clearly written, and together with the supplementary material, it provides a quite clear description of the physics. The interference between the dispersive and the dissipative coupling is responsible for interesting constructive and destructive phenomena visible in the transmission, in the modification of the mechanical behavior, and in the OMIT and OMIA spectra.

We thank the referee for the careful reading and comments of our manuscript.

However, I am not convinced that the present paper is relevant enough for the broad scientific audience of Nature Communications. In fact, the results, even though interesting, are quite technical, and they seem to me relevant only for the community working on optics and optomechanics.

We agree with the referee that the text was quite technical, which hindered its appreciation by the broad audience of Nature Communications. We believe the revised version of the manuscript improved the exposition of our results and made them more accessible for researchers outside the fields of photonics and optomechanics.

However, we respectfully disagree that our results are not relevant enough for Nature Communications' broad audience. Optomechanical devices have classical and quantum applications ranging from mass sensing, quantum transduction, and signal processing, to single-molecule detection. These go beyond the communities working on optics and optomechanics and resonate with, for example, researchers in medicine, biomedical engineering and chemistry ([doi.org/10.1038/s41565-020-0672-y](https://doi.org/10.1038/s41565-020-0672-y), [doi.org/10.1038/s41566-017-0027-x](https://doi.org/10.1038/s41566-017-0027-x), [doi.org/10.1038/ncomms12311](https://doi.org/10.1038/ncomms12311)). Furthermore, there are proposals of using these systems even in other fields of physics, as for example, in dark matter detectors ([doi.org/10.1103/PhysRevLett.126.061301](https://doi.org/10.1103/PhysRevLett.126.061301)). Dissipative optomechanics is an interesting addition to those applications, as it could be used alone or along with the dispersive coupling to further control mechanical or optical degree of freedom. We believe our results clearly demonstrate these possibilities through the interferences mentioned by the referee.

Action: The manuscript was revised and offers a simpler and more intuitive view of the phenomena.

Moreover, I am not convinced that the dissipative coupling can be controlled and improved so that it may become a useful tool for approaching the quantum regime of optomechanics, which has been already achieved with various devices using dispersive coupling. I am not sure that the relevant parameters, especially the coupling with the waveguide κ_e and the tunneling J , can be tuned so that one can have strong dissipative coupling optomechanics.

In this respect the final sentence of the abstract "The present demonstration opens a path to strongly dissipative optomechanical devices with nearly noiseless operation in the quantum regime." seems to me too strong and not justified by the results of the paper.

We appreciate the referee's comments. We agree with the referee that the present device falls short when benchmarked against state-of-the-art dispersive systems and thus have revised the tone of the claims regarding applications in the quantum regime. However, the field of dissipative optomechanics is rather young in comparison to dispersive optomechanics. The formal mathematical derivation of the phenomenon only appeared in 2009 in the manuscript by Elste et al (ref. 25). In comparison, dispersive optomechanics has been under study since the 1970's in the seminal papers of Vladimir Braginsky et al. As such, our aim in this work was demonstrating dissipative optomechanics in the sideband-resolved regime, which was elusive up to the present report. This is a first step towards using such interaction in the true quantum regime. We also stress that there are no known fundamental limits to the operation of dissipative systems in the quantum regime.

We believe our device might be used as a basis for future optimized designs, leveraging dissipative coupling for quantum applications given its unique properties, such as the strength of waveguide-cavity scattering being independent of laser-cavity detuning and the possibility of optomechanical interaction to take place even in the absence of circulating pump photons in the cavity which could cloak the optical mode against unwanted linear/nonlinear absorption.

Inspired by reviewers' 2 and 3 comments, we deepened the discussion around applications of the unambiguous interference between dissipative and dispersive couplings appearing in both mechanical and optical spectra.

Action: Revised the claims regarding a clear improvement in quantum optomechanics experiments. Added a discussion of the various uses of the dissipative and dispersive interplay in resonators and added revised figures with new informations.

A minor point is that the author could mention as possible realizations of dissipative optomechanical coupling also hybrid atom-optomechanical systems. In fact

the optical mode uncoupled to the waveguide could be replaced by an atomic system coupled only to the cavity mode. One could have similar effective dissipative coupling and interference phenomena (see for example C. Genes et al., Phys. Rev. A 80, 061803(R) (2009) and references therein.

We thank the reviewer for pointing out the references. We have included them in the revised text, along with a statement of the possibility of using atom-optomechanical systems for dissipative-like optomechanics.

In conclusion I suggest publication of the present paper in more specialized journal.

Reviewer #3 (Remarks to the Author):

- The manuscript by Primo et al. reports an optomechanical system with two coupled photonic crystal cavity modes, of which only one is coupled to an input-output channel. As a result, the displacement of a mechanical mode changes the coupling rate of the optical eigenmodes to the output channel, giving an effectively dissipative optomechanical coupling. This effect is observed in the context of transduction of thermomechanical motion, backaction, and optomechanically-induced transparency. While dissipative optomechanical coupling has been observed in other systems, the authors report in this work the first realization in a device in the resolved sideband regime.

The experimental system is state-of-the-art, the quality of the observed data is high, and the observations are clearly described by the theoretical model that includes dissipative coupling that is given in the paper. Dissipative coupling has been considered since well over a decade, but did not find wide adoption in cavity optomechanical applications. This work certainly presents an interesting experimental advance, bringing dissipative coupling to nanoscale, sideband-resolved systems. As a main motivation for the work, the authors argue that dissipative coupling could help perform quantum optomechanical experiments, in particular coherent transduction, with fewer photons resonating in the optical cavity. As in current experiments absorption from intracavity pump photons is a main limitation, this makes any significant step towards a solution to this problem one with high impact.

We appreciate the reviewer's careful reading of our manuscript.

- However, specifically on this last point I have some doubts that are not resolved in the current manuscript. I would therefore urge the authors to change the manuscript in order to address the following:

The idea that the system described in their work allows optomechanical effects of a given strength with fewer intercavity photons than for standard dispersive coupling is posed without theoretical proof or other underpinning. The idea is sketched conceptually in Fig. 1a. The accompanying text in the introduction speaks of a 'direct scattering' to the waveguide that could significantly reduce the required intracavity pump power. However, the work does not demonstrate for what parameters this advantage will become apparent. This is especially important since the dissipative coupling in this coupled-cavity system is fundamentally originating from a dispersive optomechanical coupling in one of the individual cavities. It is therefore not obvious that linear optomechanical effects and in particular coherent state transduction could be enhanced at reduced intracavity photon number.

We appreciate the reviewer's comment. The revised version of the manuscript includes an estimate of the dissipative coupling required to overcome results of state-of-the-art dispersive optomechanical systems.

Action: Included a discussion of the dissipative coupling rate required to overcome results from dispersive optomechanics experiments.

Given that the authors provide a theoretical framework, it should be straightforward to provide a calculation that supports the above claim, which forms the main motivation of the work. For instance, the authors could calculate the optomechanical cooling rate (as an example optomechanical effect) versus the intracavity photon number, for two different cases: an individual cavity with dispersive coupling G_{ω_1} and intrinsic loss rate κ_1 coupled to a waveguide at rate κ_e , and that same system optically coupled to another cavity at rate J with the same intrinsic loss rate. (Alternatively, one could also consider the depth of an OMIT dip instead of the cooling rate, for example.).

I believe the above would yield a fair comparison, in line with the studied system, that would provide a strong motivation for the presented work. I note that it would not be sufficient to instead point at the difference between the case with combined effective dispersive and dissipative coupling and the case with only the effective dispersive coupling, as observed in Figs. 4 and 5 and denoted in eq. 1: In the coupled-cavity system, both the effective dispersive and dissipative coupling to the optical supermodes are altered with respect to their values in the single-cavity system. They are thus related, and cannot be made arbitrarily large or small compared to the properties of the individual cavities in the system. In summary, one should not compare effective dispersive and dissipative coupling in a coupled-cavity system, but rather compare the dissipative coupling that emerges in the coupled system to the dispersive coupling that can be achieved with a single cavity. That is, after all, how one would choose between one or the other in an application.

If the result of the suggested comparison would show a clear and strong benefit for the coupled-cavity case, I think that could contribute to the clarity of the motivation, as well as support the discussion paragraph. If it is not unambiguous, I propose that the authors revise the claims that dissipative coupling, in this coupled-cavity system, will surely lead to a reduced power requirement for optomechanical applications such as transduction in both the abstract and the introduction.

We agree with the reviewer's comments. Indeed, one cannot generate an arbitrarily large dissipative coupling in our scheme. The main limitation arises from the adiabatic requirement of the system i.e., $2J \gg \Omega_m$. As we approach $2J = \Omega_m$, the dynamics of both optical modes become appreciable and must be accounted for, modifying the sheer dissipative coupling picture.

We do acknowledge that the present device underperforms state-of-the-art dispersive optomechanical devices. In that spirit, we have revised the claims throughout the text regarding a certain advantage of the dissipative approach over the dispersive counterpart.

Nonetheless, in this manuscript we do not make an attempt at comparing our device with state-of-the-art dispersive systems. We believe that this would render a rather unfair comparison due to the relative youth of dissipative optomechanics. For instance, OMIT and OMIA were obtained in dispersive systems over a decade ago (refs. 31 and 32), while the first dissipative optomechanical signatures in the optical spectrum are reported in the present work.

In our opinion, the main motivation of this work is showing that operating a dissipative system in the sideband-resolved regime is experimentally feasible. As acknowledged by the reviewer, such demonstration has been elusive. We believe that future work could build up on our device's design and effectively harness advantages from the dissipative coupling in the quantum regime, especially when considering that such small dissipative couplings ($g_{\kappa} = 3$ kHz) display large responses in the mechanical and optical spectra.

Inspired by reviewers' 2 and 3 comments, we deepened the discussion around applications of the unambiguous interference between dissipative and dispersive couplings appearing in both mechanical and optical spectra.

Action: Revised the claims regarding a clear improvement in quantum optomechanics experiments. Added a discussion of the various uses of the dissipative and dispersive interplay in resonators.

- Finally, I note that the authors stress the importance of achieving dissipative coupling in the resolved sideband regime. It would be helpful to the broader audience if they would discuss this need in more depth, specifically in the context of dissipative coupling. It should be noted that many theory works on dissipative coupling (e.g. ref. (26) by Elste et al. and ref. (36) by Yanay et al.) in fact stressed the ability of this mechanism to yield ground state cooling and quantum control outside the resolved sideband regime as a defining advantage, setting it apart from dispersive coupling. I suggest that the authors explain what is known in the field, i.e. that while dissipative coupling can lead to quantum control outside the resolved sideband regime (mechanical frequency smaller than total optical decay rate κ), the ratio of mechanical frequency to intrinsic optical dissipation rates should still be large. That is what is demonstrated in this work.

We agree with the referee that the broader audience would benefit from an introduction to the current state of dissipative optomechanics. We incorporated the reviewer's suggestions in the text.

Action: Added a paragraph describing the current state of dissipative optomechanics, along with the main motivations on achieving the sideband-resolved regime.

REVIEWERS' COMMENTS

Reviewer #2 (Remarks to the Author):

The new version of the paper is much more clear and the novelty is better expressed. The key result, that is, the first experimental demonstration of a dissipatively coupled optomechanical system with appreciable dispersive and dissipative coupling, at GHz frequency, is better highlighted.

However, even if the new manuscript has been significantly revised and it is improved, I am still not convinced that the paper is important enough for publication in Nature Communications. In fact, even though the optomechanical device tested here is novel, I think that the device presented here will not surpass the performance of purely dispersively coupled optomechanical system. In fact, the dissipative coupling, by construction is always smaller than the dispersive one.

Moreover, even if it is true that the modification of the mechanical dynamics is always obtained here as interference of the dispersive and dissipative coupling, I do not see how one can achieve spectacularly new physics in a regime where the dissipative coupling is always bound to be definitely smaller than the dispersive one.

Therefore, I still suggest publication of the paper in a more specialized physics journal.

Reviewer #3 (Remarks to the Author):

I have read the authors' response and revised manuscript. The manuscript was significantly revised and the claims and context were altered and specified more clearly, following the concerns I had raised previously. The clarity of the demonstrated results and their importance has strongly improved as a result. In the previous version, the authors had emphasized the possibility of dissipative coupling to enable coherent interactions at reduced intracavity photon number. I had suggested to provide theoretical underpinning for that claim. The authors chose not to perform that theoretical comparison, but instead reduced the emphasis of that advantage in the revised title, abstract, and main text, and focus on the value of the demonstrated phenomena of dynamical backaction and OMIT through dissipative interactions in a sideband-resolved, high-frequency optomechanical system. I note that the proposed claim of reduced intracavity power advantage is still mentioned in the very last paragraph. Since it is posed as an outlook rather than a motivation, the fact that it is not completely proven theoretically at this point is less problematic there in my opinion.

One could argue that the choice to remove this claim of an advantage reduced the potential impact of the current manuscript over the original. I am however of the opinion that the revisions significantly improved the manuscript. The manuscript reports new effects and new regimes associated with dissipative optomechanics with high-frequency resonators, which is of significant interest to the community and could lead to theoretical as well as experimental follow-up work. Because of that, I recommend publication of the manuscript in its current form in Nature Communications.

Response Letter: Dissipative Optomechanics in High-Frequency Nanomechanical Resonators

André G. Primo, Pedro V. Pinho, Rodrigo Benevides, Simon Gröblacher, Gustavo S. Wiederhecker, and Thiago P. Mayer Alegre.

Summary of response

We thank the referees for their once again careful review of our manuscript. Parts of the text were improved for clarity and new discussions were introduced in the Supplemental Material. In particular, we have made the following additions to the manuscript:

1. We emphasized the underlying principles leading to the dynamical backaction and optomechanically-induced transparency enhancement due to resolved-sideband dissipative optomechanical coupling.
2. The Supplemental Material was expanded to elucidate the possibility of independent control of mechanical modes using our demonstration of interference between dissipative and dispersive optomechanical couplings.

A marked-up version accompanies this rebuttal letter for easier comparison. We are confident the changes inspired by the referee's comments further highlighted the impact of our work for the broad community of Nature Communications. We once more acknowledge the referees for their invaluable input.

Reviewer #2 (Remarks to the Author):

The new version of the paper is much more clear and the novelty is better expressed. The key result, that is, the first experimental demonstration of a dissipatively coupled optomechanical system with appreciable dispersive and dissipative coupling, at GHz frequency, is better highlighted.

We appreciate the referee's careful reading of the reviewed manuscript as well as the previous comments that led to great improvement in the quality of our work and presentation.

However, even if the new manuscript has been significantly revised and it is improved, I am still not convinced that the paper is important enough for publication in Nature Communications. In fact, even though the optomechanical device tested here is novel, I think that the device presented here will not surpass the performance of purely dispersively coupled optomechanical system. In fact, the dissipative coupling, by construction is always smaller than the dispersive one.

Indeed, in our text we point out that our overall optomechanical coupling is dominated by the dispersive component of the interaction, rather than the dissipative ($|g_{\kappa_e}/g_{\omega}| = 1\%$). However, even at such small ratios, dissipative optomechanics plays a fundamental role in reshaping the interplay between acoustic and optical responses in our device, as acknowledged by the referee. This result arises from the different scaling mechanisms for dissipative and

dispersive interactions in the unprecedented resolved sideband condition explored here. While the first scales with the waveguide excitation amplitude, the latter is enhanced by the intracavity power. Remarkably, in sideband resolved systems the waveguide excitation amplitude is roughly a factor of Ω_m/κ_e larger than the intracavity power, thus leading to a large contribution from the dissipative optomechanical coupling. Consequently, even though the referee's comment is accurate for the presented device, there's no known *fundamental* limit for how well dissipative optomechanical devices could perform when benchmarked with its dispersive counterparts, specially if one considers how the waveguide-field enhancement could be leveraged to increase the phenomena already demonstrated in our manuscript.

We would also like to emphasize that this work paves the way for future optimized designs and theoretical work, which could be further extended to distinct optomechanical systems, e.g., electromechanical resonators and exciton-polaritons (Refs. 47 and 25). In fact, ours is the first demonstration of dissipative optomechanics in the sideband-resolved regime, a feature that was demonstrated more than 15 years ago for dispersive optomechanics in integrated devices (Schliesser, A., *et al. Nature Phys* 4, 415–419, 2008). This is the required regime for any future scalable quantum applications of optomechanical systems.

Action taken: *Expanded discussion and conclusion to further highlight the mechanism of enhancement of the dissipative optomechanics leading to appreciable signatures in the data despite the reduced dissipative coupling rate.*

Moreover, even if it is true that the modification of the mechanical dynamics is always obtained here as interference of the dispersive and dissipative coupling, I do not see how one can achieve spectacularly new physics in a regime where the dissipative coupling is always bound to be definitely smaller than the dispersive one. Therefore, I still suggest publication of the paper in a more specialized physics journal.

We respectfully disagree with the referee. We reiterate the key novelties demonstrated here that make this article worth publishing in this prestigious journal:

- The selective addressing of the mechanical modes demonstrated in our manuscript. As also acknowledged by referee 3, this is a new regime arising from the interference between dissipative and dispersive optomechanics. We stress that purely dispersive optomechanical systems do not display this kind of phenomena. As mentioned in the concluding remarks in the main text, this interplay could be explored for classical and quantum applications, such as the generation of mechanical exceptional points and individual addressing of qubits, respectively.. To further substantiate our claims and emphasize our novelty, we provide further discussion (S7 of the updated Supplementary Material) in showing that individual control of mechanical modes is achieved at reasonably low dissipative-dispersive coupling ratios, e.g $|g_{\kappa_e}/g_{\omega}| = 5\%$, provided the system is in the sideband-resolved regime.

- The concepts and underlying mechanisms for creating the dissipative coupling presented in our work could be readily adapted even for systems in the sideband-unresolved regime ($\Omega_m \ll \kappa$) while leveraging the dissipative coupling to achieve ground-state cooling. As noted by referee 3 in the first round of reviews, this is known to be unfeasible using dispersive optomechanics alone (Refs. 25, 26, 27, 28 and 29 of the main text).

Action taken: The Supplementary Material was updated to include results displaying required conditions for the complete individual control of mechanical modes in our system.

Reviewer #3 (Remarks to the Author):

I have read the authors' response and revised manuscript. The manuscript was significantly revised and the claims and context were altered and specified more clearly, following the concerns I had raised previously. The clarity of the demonstrated results and their importance has strongly improved as a result. In the previous version, the authors had emphasized the possibility of dissipative coupling to enable coherent interactions at reduced intracavity photon number. I had suggested to provide theoretical underpinning for that claim. The authors chose not to perform that theoretical comparison, but instead reduced the emphasis of that advantage in the revised title, abstract, and main text, and focus on the value of the demonstrated phenomena of dynamical backaction and OMIT through dissipative interactions in a sideband-resolved, high-frequency optomechanical system. I note that the proposed claim of reduced intracavity power advantage is still mentioned in the very last paragraph. Since it is posed as an outlook rather than a motivation, the fact that it is not completely proven theoretically at this point is less problematic there in my opinion.

One could argue that the choice to remove this claim of an advantage reduced the potential impact of the current manuscript over the original. I am however of the opinion that the revisions significantly improved the manuscript. The manuscript reports new effects and new regimes associated with dissipative optomechanics with high-frequency resonators, which is of significant interest to the community and could lead to theoretical as well as experimental follow-up work. Because of that, I recommend publication of the manuscript in its current form in Nature Communications.

We are glad for the referee's appreciation of our work and we are pleased to hear that the revisions we made have addressed the concerns you raised and improved the clarity and significance of the results. The effects of a possible intracavity power reduction are currently under exploration in our group and will be the central theme of a future manuscript.